# Heat Shock Proteins Alterations in Rheumatoid Arthritis

**DOI:** 10.3390/ijms23052806

**Published:** 2022-03-03

**Authors:** Malak Fouani, Charbel A. Basset, Giuseppe D. Mangano, Lavinia G. Leone, Nada B. Lawand, Angelo Leone, Rosario Barone

**Affiliations:** 1Department of Biomedicine, Neuroscience and Advanced Diagnostics (BIND), University of Palermo, 90127 Palermo, Italy; malak.fouani@unipa.it (M.F.); charbel.basset@unipa.it (C.A.B.); giuseppedonato.mangano@unipa.it (G.D.M.); angelo.leone@unipa.it (A.L.); 2School of Medicine, University of Palermo, 90127 Palermo, Italy; laviniagleone@gmail.com; 3Department of Anatomy, Cell Biology and Physiological Sciences, Faculty of Medicine, American University of Beirut, Beirut 1107-2020, Lebanon; nl08@aub.edu.lb; 4Department of Neurology, Faculty of Medicine, American University of Beirut, Beirut 1107-2020, Lebanon

**Keywords:** rheumatoid arthritis, heat shock proteins, inflammation, neurogenic inflammation, vaccine, HSP therapy

## Abstract

Rheumatoid arthritis (RA) is a chronic inflammatory and autoimmune disease characterized by the attack of the immune system on the body’s healthy joint lining and degeneration of articular structures. This disease involves an increased release of inflammatory mediators in the affected joint that sensitize sensory neurons and create a positive feedback loop to further enhance their release. Among these mediators, the cytokines and neuropeptides are responsible for the crippling pain and the persistent neurogenic inflammation associated with RA. More importantly, specific proteins released either centrally or peripherally have been shown to play opposing roles in the pathogenesis of this disease: an inflammatory role that mediates and increases the severity of inflammatory response and/or an anti-inflammatory and protective role that modulates the process of inflammation. In this review, we will shed light on the neuroimmune function of different members of the heat shock protein (HSPs) family and the complex manifold actions that they exert during the course of RA. Specifically, we will focus our discussion on the duality in the mechanism of action of Hsp27, Hsp60, Hsp70, and Hsp90.

## 1. Introduction

Rheumatoid arthritis (RA) is a chronic inflammatory autoimmune disease, where the immune system attacks the joints causing, sustained inflammation. RA affects approximately 3 individuals in a 10,000-person population globally each year. Hence, the prevalence rate is projected to be around 1%; nonetheless, it increases with age and typically peaks between 35 and 50 years [1]. Multiple other risk factors affect the onset of the disease, such as sex and place of residence, as women and inhabitants of developed countries are more prone to RA [1]. Additionally, the type and pattern of joint involvement affect the initiation of RA. Moreover, the progress of this disease may also be different according to several variables, including the frequency of joint swelling, genetic predisposition, and the presence of autoimmune antibodies in the serum. RA progression is classically divided into four stages [2]: (i) the immune system misguidedly attacks joint tissue, causing swelling and inflammation of the synovium (stage 1); (ii) the swelling worsens and the connective tissue surrounding the joint becomes damaged (stage 2); (iii) the characteristic symptoms of inflammation are more severe; and screening tests become less significant for diagnosis as the physical deformity in patients’ extremities is distinctive (stage 3); (iv) the joints and cartilage are completely destroyed and the bones become fused together (stage 4). While progression through these stages takes a considerable number of years, the patients persistently suffer from tenderness, joint stiffness, swollen joints, fever, fatigue, and loss of appetite. Accordingly, at the outset of the disease, small joints are mostly affected [2]. However, as the disease progresses, symptoms often extend to the knees, wrists, elbows, ankles, shoulders, and hips. Besides, about 40% of the people who have rheumatoid arthritis also experience signs and symptoms that don’t involve the joints such as in the skin, eyes, and kidneys, especially at later stages [3].

## 2. Pathogenesis of Rheumatoid Arthritis

In RA, the immune system, which is naturally responsible for defending the body against foreign invaders, turns on the cells and proteins in the joints’ lining and destroys them [1]. This autoimmune action compromises the integrity of cartilage, bone, and the lining of the joint cavity, named synovium, which secretes a fluid allowing a smoother movement. Subsequently, a rheumatoid arthritis joint emerges and can be easily discerned from its healthy counterparts through the following features: bone erosion, degraded cartilage, and inflamed synovium [4]. These structural deformities consequently lead to increased friction and damage to the joint. Moreover, blood vessels in the inflamed joint become leaky, leading to its invasion by activated lymphocytes (T and B), macrophages, and neutrophils. This infiltration plays a role in the autoimmune processes underlying this disease and the progression of the inflammatory response [5]. B cells function as one of the underlying factors of RA onset by secreting chief proteins such as rheumatoid factors (RFs), anti-citrullinated protein antibodies (ACPA), and proinflammatory cytokines. B cells also mediate T cell activation through the synthesis of costimulatory molecules [6]. T cells’ main function in RA is the activation of macrophages and fibroblasts and their subsequent conversion into tissue-destructive cells. Furthermore, macrophages play a parallel role to T cells and B cells, by producing a diversity of cytokines and chemokines that engender and sustain the inflammation [6].

## 3. Neurogenic Inflammation and Rheumatoid Arthritis

Several mechanisms were proposed to explain the pathophysiological process of this autoimmune disorder; chief among them is the intricate interaction between the nervous system and the immune cells. In response to the auto-immune activity in RA, the resident immune cells are activated leading to peripheral sensitization. Therefore, an integrated network is formed via the synthesis and release of inflammatory mediators, which interact with neurons, glia, and other immune cells in order to synchronize the immune responses and regulate the excitability of pain pathways. The local inflammatory reaction and resulting secreted intermediates can sensitize sensory neurons and lead to the generation of dorsal root reflexes, a phenomenon responsible for the initiation of neurogenic inflammation and the resultant hyperalgesia [7]. Peripheral small-diameter primary afferent fibers originating from the dorsal root ganglion, specifically Aα, Aβ, and C fibers are the chief nerve fibers involved in this process. Aα fibers specialize in mechanosensation, while Aβ and C fibers’ function pertains to nociception.

These neuronal extensions produce glutamate, substance P (SP), and calcitonin gene-related peptide (CGRP) that contribute to the symptoms of neurogenic inflammation. Peripherally, these substances act on post-capillary venules, rendering them leaky, resulting in plasma extravasation and vasodilatation [8]. SP has shown a wide range of effects on the cellular components of the peripheral inflammatory cascade. Moreover, SP is involved in pain transmission within the central nervous system, where it appears to modulate widespread and prolonged changes in the sensitivity of dorsal horn neurons to various stimuli. Furthermore, CGRP alongside various molecules, such as bradykinin and prostaglandins as well as a multitude of proinflammatory cytokines (interleukins 1, 6, 17, nerve growth factor beta (NGF-β), and tumor necrosis factor alpha (TNF-α)), which directly alter the responses of nociceptive neurons, have been found abundantly in the synovial fluid of RA patients.

As for glutamate, synovial fluid (SF) concentrations of this molecule have been shown to significantly increase in arthritis animal models and have been correlated with a rise in inflammatory mediators. This neurotransmitter is fundamental in pain transmission and is now known to be crucial in the signaling in various non-excitable cells, being released by macrophages, lymphocytes, synoviocytes, osteoblasts, osteoclasts, and chondrocytes, acting on ionotropic glutamate receptors (iGluRs) and metabotropic GluRs in multiple joint cell types.

All these interactions contribute to peripheral sensitization and hyperexcitability of nociceptive neurons in the central nervous system. This convoluted network between the nervous system and the immune cells plays an essential role in the initiation and continuance of chronic pain in this autoimmune disease (Figure 1).

## 4. Rheumatoid Arthritis and a Specific Pool of Immunogenic Proteins

RA is a complex autoimmune disease and its etiology has always been the subject of investigation. Multiple factors, including genetic makeup, environmental triggers, hormonal imbalance, and specific proteins’ expression, were shown to contribute to the progression of the disease. In addition, a series of inflammatory markers and immunogenic proteins involved in the underlying mechanism of the autoimmune response and persistent inflammation have been identified [6]. Among these are the C-reactive proteins, interleukins, and circulating P-selectins, which are released from specific immune cells to sensitize nociceptors, while TNF-alpha, interleukin-15, and other neuropeptides, secreted by sensory neurons, interact with specific receptors on immune cells and cause further sensitization [9,10].

Within the immune system, B cells, T cells, and macrophages are the three chief cells that contribute to the progression of RA [11]. Disrupted immune tolerance at the cellular and non-cellular levels is a key factor in the autoimmune inflammatory process. Several studies have demonstrated a significant increase in autoreactive mature naïve B-cells in the blood of untreated RA patients versus healthy ones. These localized cells stimulate an increase in major inflammatory cytokines, such as TNF-α, IL-6, IL-12, IL-23, and IL-1α, in the affected tissue [6]. In addition, it has been validated that memory B cells from peripheral blood, synovial fluid, and tissues of RA patients showed an increased production of receptor activator of nuclear factor κB ligand (RANKL) [6]. RANKL, an important cytokine, binds to its receptor and mediates bone resorption through the differentiation and activation of osteoclasts. B cells also secrete physiologically important proteins in RA, such as rheumatoid factors (RFs), and anti-citrullinated protein antibodies (ACPA). Interestingly, in RA patients, T cells are activated by both B cells and macrophages. However, its precise role in the development of RA remains to be unraveled. A profile of cytokines expressed by synovial T cells, dominated by IL-2, IL-4, IL-13, IL-17, IL-15, basic fibroblast growth factor, and epidermal growth factor, plays a major role in the pathogenesis and progression of the disease [6]. In the early course of the disease, macrophages can also play a critical role in T cell activation resulting in the production of effector T cells and increased expression of proinflammatory mediators, such as IL-1α, IL-1β, and MMPs [12].

Therefore, competing targeted therapeutics, such as anti-CD3, CTLA4-Ig, anti-TNF-α, or B cell–depleting antibodies, have been used for the suppression of the inflammatory response [13,14]. Nevertheless, they do not alter immune tolerance. As a result, several therapeutic approaches have shifted focus to finding novel ways to dampen this inflammation, restore immunotolerance, and re-establish physiological regulation. In this context, effort has been made to target heat shock proteins (Hsp27, Hsp60, Hsp70, and Hsp90), a family of molecular chaperones that participate in immune response as part of their noncanonical function, as they have been found to become highly expressed during inflammation. As the name implies, these proteins are secreted by cells that are exposed to elevated temperatures above the physiological norm; however, they can also be induced by toxins, oxidants, and stressors [15]. Due to their antioxidant effect and amphitropic characteristics, HSPs expressed in microglia, astrocytes, and endothelial cells are crucial to many neurological disorders and play an important role in protecting nerve cells from injury [16].

As intracellular molecular chaperones, HSPs ensure the correct folding of both normal and damaged proteins [17]. Additionally, they are linked to peptide binding to MHC molecules (also known as antigen presentation), protein synthesis, and transport. HSPs can also aid in protein refolding and translocation through the membranes and the dissolution of protein aggregates. Fundamentally, they ensure the preservation of protein homeostasis [18,19]. HSP members are classified according to their molecular weight [20], and classified as families of phylogenetically related proteins, such as the Hsp90, Hsp70, Hsp40, and HSP—crystallin families. Other HSPs might have molecular weights within the postulated ranges but are not classified as members of those families [21]. As members of the molecular chaperone family, HSPs can be localized in all cells, cell compartments, tissues, and outside cells in the intercellular space, in circulation, and in secretions [19]. The term chaperonins usually refers to the chaperones with a molecular weight in the range 55–64 kDa and that have been classified into Group I and II. They are composed of oligomeric double-ring protein assemblies that postulate vital kinetic assistance to protein folding by binding non-native proteins [19,22].

In RA, HSPs have been shown to possess immunoregulatory attributes for their immunogenic trait [23], particularly Hsp27, 60, 70 and 90. Many studies have demonstrated their role in dampening the inflammation, lessening the severity of the disease, and delaying its onset. Suppression of arthritis by HSPs is attributable to increased release of the anti-inflammatory cytokine, IL-10, and subsequent decrease in T cell production. This cascade of biological responses greatly affects the secretion of inflammatory cytokines and alters the inflammatory process. Inarguably, the interplay between the HSPs and the immune system is very complex; these immunogenic proteins, under a range of contexts, have the ability to play a dual role: a stimulatory and an immunoregulatory role. In the following sections, we will provide a detailed insight into the behavior of specific HSPs and their contribution to the development and progression of RA.

## 5. Rheumatoid Arthritis and Hsp27

Heat shock protein 27 (Hsp27), also known as heat shock protein beta-1 (HSPB1), belongs to the HSP family characterized by its small molecular weight (12–43 kDa) [24]. Hsp27 and other members of the small HSP family share a preserved c-terminal domain, the α-crystallin domain, which is identical to the vertebrate eye lens α-crystallin. This protein plays a pivotal role in multiple pathologies, such as cancer, as well as neurodegenerative, cardiovascular, and autoimmune diseases such as RA [24].

Hsp27 displays diverse inflammatory effects, i.e., exogenous Hsp27 can prevent neutrophil apoptosis in a dose-dependent manner without altering the levels of immuno-inflammatory cytokines, such as IL-12 and IL-10. This variation prolongs the survival of neutrophils and consequently exacerbates acute inflammation and tissue destruction [25]. The proinflammatory effect of Hsp27 is also evident in a study that investigated the role of small HSPs and neurovascular inflammation in Alzheimer’s disease. The results showed that Hsp27 increased the production of IL-8, a pro-inflammatory cytokine, and dampened the anti-inflammatory response by decreasing the production of transforming growth factor β1 (TGF-β1) in smooth muscle cells and astrocytes [26], showing Hsp27′s ability to enhance neurogenic inflammation. Additionally, Hsp27 is essential for the activation of IL-1 and TNF upstream signaling pathways that are part of the p38 mitogen-activated protein kinase (MAPK) inflammatory response, which is a fundamental process in synovial inflammation [27]. Therefore, even though research on the proinflammatory role of Hsp27 in the progression of RA is missing, a possible link is worth investigating.

A study did, in fact, demonstrate two members of the sHSP family, αA crystallin and HSPB8, were able to activate dendritic cells by inducing maturation and inflammatory cytokine production. Therefore, they can be assumed to be rigorously involved in the pathogenesis of RA. Though, Hsp27 itself did not show any significant changes. Nonetheless, it has been confirmed that there is a significant upregulation of Hsp27 mRNA in RA synovial tissues, suggesting its involvement in the peripheral mechanisms associated with RA [28]. Conversely, evidence has demonstrated that exogenous Hsp27 is a powerful inducer of human monocyte-derived IL-10 which is characterized by its anti-inflammatory properties [29]. Additionally, a study showcased the ability of Hsp27 to induce the release of inhibitory mediators, such as thrombospondin-1 and IL-10, and inhibit the differentiation of monocytes into immature dendritic cells or macrophages [30]. This leads to a decrease in T cell activation, a protective mechanism in autoimmune diseases, such as RA. Furthermore, HSPB1 was shown to be able to inhibit the activation of the nuclear factor kappa B (NF-κB) pathway and the subsequent decrease in TNF-α secretion in microglial cells, which could showcase its role in dampening neurogenic inflammation [16].

Therefore, Hsp27 induction could be a double-edged sword in the context of RA development (Figure 2).

## 6. Rheumatoid Arthritis and Hsp60

Heat shock protein 60 (Hsp60) belongs to the family of HSPs discerned by their 60 kDa molecular mass [23]. They function within the cell as chaperones to aid in the proper folding of nascent polypeptides and confer cytoprotection during stressful situations [23,31]. Hsp60 has been highly recognized in autoimmune and inflammatory processes as an immunogenic protein reflecting the states of cells and tissues in certain pathophysiological diseases, such as RA [32]. It provides the immune system with reliable biomarkers and helps determine the proinflammatory or anti-inflammatory outcome through its molecular concentration, its precise peptide moiety, and the cell types that it activates [33]. Hsp60 affects components of both innate and adaptive immunity. For the former, Hsp60 of different sources can trigger monocyte differentiation into macrophages and enable the secretion of various proinflammatory cytokines [34]. Meanwhile, for the latter, this protein proved to be essential for the development and maintenance of regulatory T cells that are positive for CD4, CD25, and Foxp3 [35].

The proinflammatory role of Hsp60 is highlighted in multiple studies. First, it possesses the ability to exacerbate IL-Iβ-induced inflammation in microglia via the TLR4-p38 MAPK pathway, highlighting its function in empowering neurogenic inflammation [36]. This concept is reinforced by a study that suggested that Hsp60 plays a proinflammatory role in diabetes-induced neuroinflammation by stimulating pattern recognition receptors across neurons and astrocytes to propagate the inflammation [37]. Moreover, extracellular Hsp60 binds the microglial lectin-like oxidized low-density lipoprotein receptor-1(LOX-1) receptor and induces the consequent production of proinflammatory cytokines [38].

Besides, in patients with type 2 diabetes, Hsp60 can possibly interact with toll-like receptors found on vascular endothelial cells to stimulate the release of proinflammatory cytokines [39]. Furthermore, Hsp60 is implicated in adipose tissue inflammation by activating proinflammatory signaling cascade in skeletal muscle cells and adipocytes [40]. More specifically, increased Hsp60 release in adipocytes stimulates the production of proinflammatory cytokines, such as TNFα, IL-6, and 8 while, in skeletal muscle cells, it activates extracellular signal-related kinase (ERK)-1/2, Jun NH (2)-terminal kinase (JNK), and nuclear factor (NF)-κB [40]. ERK and JNK induce inflammatory activation of stromal fibroblast-like synoviocytes (FLS) derived from synovial tissue, chondrocytes and osteoclasts in patients with RA [41]. NF-κB is involved in RA pathology, through the development of T helper 1 responses, activation, abnormal apoptosis and proliferation of RA fibroblast-like synovial cells, and differentiation and activation of bone resorbing activity of osteoclasts [42]. This is supported by another study, which showed that recombinant *Wolbachia* Hsp60 acts on monocytes to stimulate the gene expression of proinflammatory cytokines, such as TNF-α, IL-1β, and IL-6 [43]. Interestingly, Hsp60 also induced apoptosis in monocytes but not in lymphocytes, suggesting a role in inflammation-mediated monocyte disruption [43]. Given the proinflammatory properties of Hsp60, addressing its potential implication in RA is warranted.

Thus, multiple research groups directed their research towards targeting Hsp60 family members in an effort to discover novel therapies for RA. For example, one investigated the impact of the preimmunization with Mycobacterial 65-kDa Hsp in male rats affected with adjuvant-induced arthritis (AIA). This study was conducted to show the protective role of this preimmunization in AIA and to specify the epitopes recognized by rat T cells. T cells play a pivotal role in the underlying inflammatory and anti-inflammatory pathways of this disease. Furthermore, cross-reactive immunogenic recognition of mycobacterial Hsp65 and endogenous self-Hsp60 at the T cell level was detected, which, consequently, emphasized the protective role of Hsp60 preimmunization in dampening the inflammation and decreasing the severity of the disease [44]. The same hypothesis was presented in another study that showed for the first time that rats can be protected from AIA by immunization with naked DNA encoding for mycobacterial Hsp65. An interesting finding in this study was that the intracellular expression of Hsp65 allowed the entrance to the cell’s MHC class I and class II presentation pathways and, subsequently, stimulated CD4+ and CD8+ cells [45].

In contrast, other data point towards a protective role for Hsp60 in RA via enabling the secretion of regulatory anti-inflammatory cytokines, IL-4 and IL-10. Consequently, Hsp60 results in the dampening of the RA-related inflammation [46]. Moreover, it has been shown that nasal administration of a peptide analog of an arthritis-related Hsp60 T cell epitope resulted in the inhibition of AIA. This obstruction was largely due to the production of regulatory cells that produce the anti-inflammatory cytokines TGF beta, IL-4, and IL-10. IL-10 specifically induced autoimmune tolerance [47]. Most importantly, Hsp60 was shown to induce a tolerogenic immune response in several models of RA, resulting in the upregulation of IL-10, an anti-inflammatory and immune-suppressive cytokine that promotes interaction of T cells with other immune cells, thus leading to the inhibition of RA [46].

Considering these findings combined, Hsp60 family members might play a dual role when it comes to RA (Figure 3).

## 7. Rheumatoid Arthritis and Hsp70

Members of the 70-kDa heat shock protein (Hsp70) family, as with all molecular chaperone families, are well characterized and highly evolutionarily preserved [48]. They are vastly upregulated in the cell in response to the stress of various origins. Hsp70 proteins have three highly conserved functional domains: a substrate-binding domain, whose function is still poorly understood, as well as N-terminal ATPase and C-terminal domains, which participate in the folding of nascent proteins under both physiological and stressful conditions. Hsp70 have been shown to possess an influence on the innate and adaptive immune responses, as well as autoimmune reactions [49].

Multiple roles for Hsp70 have been recorded in the context of RA. For instance, it was suggested that extracellular human Hsp70 may possess a proinflammatory impact that contributes to the progression of RA [50]. This potentiating activity of Hsp70 might be due to the rise in both T helper 17(Th17) cell frequencies and Th17/Treg ratio, as such actions were seen in mononuclear cell cultures, possibly through increasing IL-6 production [50]. However, this notion was questioned, as the increased levels of Hsp70 in RA patients did not correlate with RA severity. Nonetheless, the involvement of Hsp70 in RA commencement and development was supported by different research groups. For instance, Kang et al. showed that the knockout of this protein in RA fibroblast-like synoviocytes protected these cells from nitric-oxide-induced apoptosis via the Akt signaling pathway [51]. Moreover, Mantej and colleagues proved that the levels of anti-Hsp70 autoantibodies are significantly higher in sera of RA patients when compared to those of normal controls [52]. Additionally, this increase in anti-Hsp70 IgM levels was significantly correlated with a decrease in the amounts of the proinflammatory TNF-α in RA patients, thus leading to the dampening of the inflammatory response [52]. Furthermore, another study demonstrated that DNA vaccination by Hsp70 inhibits AIA in rat models. This impeachment of the disease occurred due to the shift of the arthritogenic T cells from the proinflammatory Th1 phenotype to the Th2/3 phenotype, which lead to the subsequent changes in their cytokine secretion profiles from an interferon- γ (IFNγ) dominated scene to one where IL-10 and TGFβ1 prevailed [53]. The potential involvement of Hsp70 in neurogenic inflammation could be through inducing the secretion of TNF-α and IL-6 in microglia [54].

Thus, the proapoptotic and possible proinflammatory properties of Hsp70 encourage further exploration of its link to RA.

In contrast, Hsp70 has also been affiliated with anti-inflammatory properties in RA. Extracellular Hsp70 specifically has been shown to down-regulate the inflammation process in preclinical models of rheumatoid arthritis through the modulation of several signaling pathways. The most notably modified path is the nuclear factor kappa B (NF-κB) signaling pathway, whose inhibition by Hsp70 leads to the downregulation of inflammatory cytokines, such as IL-6, IL-8, and monocyte chemoattractant protein-1 (MCP-1). Interestingly, these results were established through the usage of extracellular Hsp70 obtained from synoviocyte cultures of rheumatoid patients [55]. This anti-inflammatory role of Hsp70 is corroborated by other experimental studies that have described the importance of Hsp70 in directing the development of T cells into regulatory T cells (Tregs) rather than into proinflammatory Th1 cells [53]. Concomitantly, in a clinical setting, patients with RA who received a single intravenous infusion of the binding immunoglobulin protein (BiP), a protein belonging to the Hsp70 family, showed remission of symptoms. BiPs had a pivotal role in inhibiting the severity of the disease through deactivation of human monocytes and abolition of dendritic cell maturation, leading to the production of CTLA-4+ regulatory T cells [56]. Thus, a potential anti-inflammatory activity is possible for Hsp70 in the context of RA.

On another note, an Hsp70 family member named Hsc70 proved to be able to inhibit the transient receptor potential vanilloid type 1 (TRPV1) channel at the level of the dorsal root neurons, including the small nociceptors via ROCK activation. This decreased the pain sensation in mice injected with Complete Freund’s Adjuvant, an inflammatory agent known to induce RA. This study showcased the role of Hsp70 family members in controlling not only RA, but the associated neurogenic inflammation. In addition, it has been shown that the overexpression of Hsp70 exerted an anti-inflammatory effect in the brain of mice [57].

Considering the above, Hsp70 may play a selective role in the immunoregulatory process of RA (Figure 3).

## 8. Rheumatoid Arthritis and Hsp90

The HSPs belonging to the family of 90 kDa chaperones (Hsp90) are one of the most common HSPs. Aside from their classical roles in proper protein folding, protein stabilization against heat stress, and aiding in protein degradation, Hsp90 expression and activity is increased in cancer cells to enhance the function of several oncoproteins [58].

Hsp90 was shown to increase in the sera of RA patients compared to normal controls [52]. Hsp90 family members can also be considered as possible therapeutic targets in RA due to their role in activating the humoral immune response by inducing the production of auto-Hsp antibodies [59]. Thus, Hsp90 molecular target inhibition has been investigated as target therapy for patients with RA [60]. For example, SNX-7081 has been shown to inhibit NF-kB nuclear translocation and the consequent inflammatory cytokine and nitric oxide release [61]. Moreover, it markedly prevented ankle and knee swelling, as well as cartilage or bone disruption in two animal models of arthritis that included rat collagen-induced arthritis (CIA) and adjuvant-induced arthritis (AIA) [61]. Another Hsp90 inhibitor, EC144, blocked the adaptive immune response by inhibiting the activation of CD4+ T cells and consequently suppressed RA progression in rat CIA [62].

Furthermore, the decrease in Hsp90 activity due to antibody binding has also been studied in the context of RA therapy. For instance, the vaccination of adjuvant arthritic rats with human Hsp90 DNA can arrest the progression of the disease. This occurred through increased T cell secretion of IL-10 and TGFβ1 [53]. Furthermore, Liu and colleagues have shown that immunization with Hsp90 stimulates the development of Treg cells, leading to the suppression of cytotoxic T lymphocytes. Their results illustrate the immunomodulatory role of Hsp90 and its capability of maintaining self-tolerance [63].

Despite its role in the pathophysiology of RA, a discrepancy in the data emerges, as shown by a clinical study, where it was shown that anti-Hsp90 antibodies increase in RA patients and correlate with an increase in the secretion of IFN-ɣ for anti-Hsp90 IgG and rheumatoid factor for anti-Hsp90 IgA [52]. The collective data indicate that Hsp90 appears to be a promising protein for further investigations and for molecular-based therapies in the context of RA.

Based on the published data, we propose a model depicting the signaling pathways by which Hsp60, 70, and 90 are involved in the pathology of RA (Figure 3).

## 9. Conclusions

Recent studies investigating the mechanisms underlying the development and progression of RA have highlighted the crucial role of specific members of the HSP family. However, whether HSPs are directly involved in the pathology of the disease or have an indirect modulatory function remains to be determined. In this review, we discuss the anti-inflammatory and immunomodulatory role of Hsp27, Hsp60, Hsp70, and Hsp90 in the context of RA, bearing in mind that this field is still under study. We have also touched upon some of the specific immunoregulatory pathways and receptors responsible for the alteration of HSP expression in inflamed tissues. Nevertheless, the inflammatory role of those specific HSPs were also tackled, showing the inflammatory pathways responsible for the pathogenesis of this autoimmune disease. Since HSPs have been implicated in a variety of chronic inflammatory conditions, understanding the course and mechanism of action of these proteins specifically in RA will be crucial to determine their potential as a target for therapeutic intervention, in which, if HSPs were playing an immunomodulatory role, they themselves could be a pharmacological treatment, or compounds regulating the heat-shock response could be a promising focus in chronic autoimmune diseases. Conversely, if these proteins are playing an inflammatory role, therapeutic intrusions to fine-tune the amount of HSPs expressed or inhibit the inflammatory pathways of this compound could possibly open up new avenues for inflammation regulation.

## Figures and Tables

**Figure 1 ijms-23-02806-f001:**
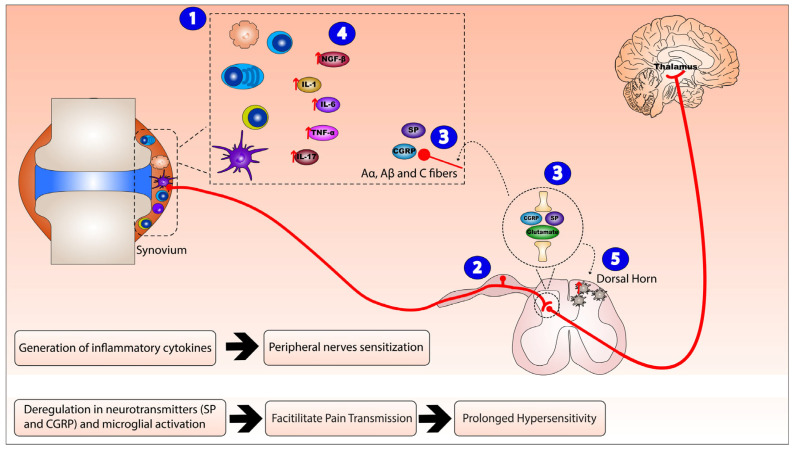
Neurogenic inflammation mechanism of action in Rheumatoid arthritis (RA). The mechanism of action of neuroinflammation in RA consists of: (1) activation and invasion of inflammatory synovial cells (T cells, B cells, macrophages, plasma cells, and dendritic cells) leading to peripheral sensitization and local inflammatory reaction that (2) sensitizes peripheral small diameter fibers (A, B, and C fibers), hence generating dorsal root reflexes. (3) Peripheral nerve endings and the central nerve endings originating from the dorsal root ganglion release substance P (SP), calcitonin gene-related peptide (CGRP), and glutamate that peripherally act on post-capillary venules, rendering them leaky and resulting in plasma extravasation and vasodilation, while, centrally, they trigger neurogenic inflammation. Therefore, this integrated inflammatory network (4) enhances the release of inflammatory cytokines nerve growth factor beta (NGF-β), interleukin (IL)-1, IL-6, IL-17, and tumor necrosis factor alpha (TNF-α) (as indicated by the red arrows) and (5) activates microglial cells in the dorsal horn of the spinal cord.

**Figure 2 ijms-23-02806-f002:**
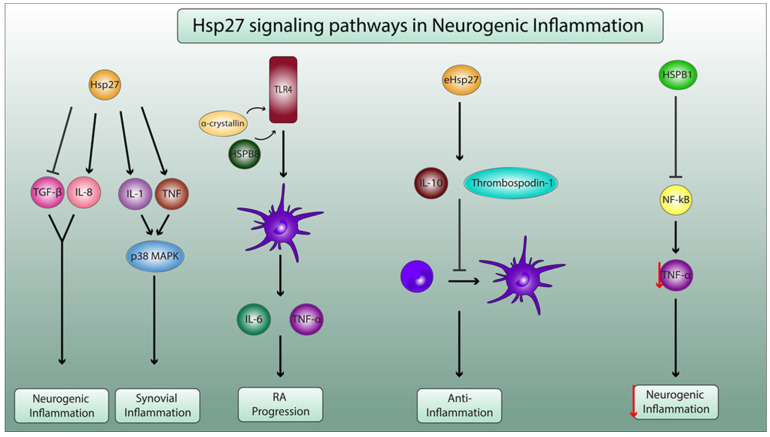
Molecular mechanisms of Hsp27 in neurogenic inflammation and in pathophysiology of RA. Hsp27 enhances the production of the proinflammatory IL-8 and decreases the secretion of anti-inflammatory transforming growth factor β1 (TGF-β1), leading to neurogenic inflammation. Additionally, Hsp27 increases the secretion of IL-1 and TNF, which activate the p38 mitogen-activated protein kinase (MAPK) pathway, leading to synovial inflammation. α-crystallin and HSPB8 activate toll-like receptor 4 (TLR4), thus enhancing dendritic cell maturation and proinflammatory cytokine secretion (IL-6 and TNF-α), leading to RA progression. Exogenous Hsp27 (eHsp27) enhances the production of inhibitory mediators thrombospodin-1 and IL10, leading to the decrease in the differentiation of monocytes to dendritic cells, ensuring anti-inflammatory properties. HSB1 inhibits the activation the pf nuclear factor kappa B (NF-κB) pathway and consequently decreases the secretion of TNF-α (as indicated by the red arrow) by microglial cells, thus playing a pivotal role in dampening neurogenic inflammation.

**Figure 3 ijms-23-02806-f003:**
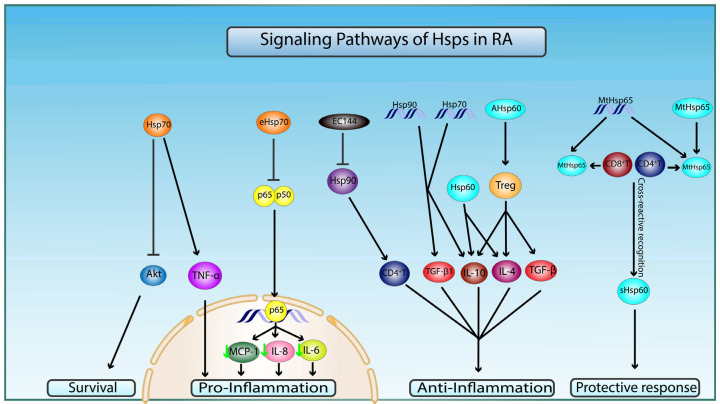
Molecular mechanisms of Hsp60, 70, and 90 in pathophysiology of RA. Preimmunization with naked DNA of mycobacterial Hsp65 (MtHsp65) or MtHsp65 leads to recognition of Hsp65 and cross-reactive recognition of self-Hsp60 (sHsp60) epitopes by CD4+ and CD8+T cells, which confer protection in RA through a T-cell-mediated protective immune response. Intracellular Hsp60 and administration of arthritis-related Hsp60 T-cell epitope (AHsp60) can dampen RA-related inflammation by T-regulatory cells (Treg)-mediated stimulation of secretion of anti-inflammatory cytokines TGF-β, IL-4, and IL-10. Hsp70 increases proinflammation and apoptosis by upregulating TNF-α and inhibiting pro-survival Akt signaling pathways, respectively. Hsp70 and Hsp90 DNA vaccination conferred anti-inflammatory protection via increase in anti-inflammatory cytokines TGF-β1 and IL-10. Extracellular Hsp70 (eHsp70) downregulates inflammatory processes by inhibiting nuclear translocation of NF-κB (p65/50) and transcription of proinflammatory cytokines MCP-1, IL-6, and IL-8 (green arrow indicates a decrease in gene expression). Target inhibition of Hsp90 by EC144 promotes the diminishment of CD4+T-cell-mediated immune response.

## Data Availability

Not applicable.

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
