# Peer review of "Heat Shock Proteins Alterations in Rheumatoid Arthritis"

_ijms, 2022, doi:10.3390/ijms23052806_

Round 1

Reviewer 1 Report

The authors reported HSP contribution in RA pathogenesis.

The article was interesting and useful to understand the roles of HSP in RA, however, the followings are needed to publish in the Journal.

  1. I feel strange the expression of 'the body's' in the text.
  2. I hardly understand 'neurogenic inflammation'. Do you have another term instead of 'neurogenic'?
  3. Line 57: named synovium and Line 60: synovial membrane   These gave me unfamiliar meaning.

Author Response

Subject: Manuscript ID: ijms-1621218 “Heat Shock Proteins Alterations in Rheumatoid Arthritis”

Dear Editor,

Thank you for giving us the opportunity to resubmit our manuscript after the peer review process. The comments of the reviewers were very appropriate and we modified the text of the manuscript and the figures in accordance with the reviewers’ requests. All new or modified parts are visible in red font. Below this message you will find the response to the reviewers’ comments.

We hope the manuscript is now acceptable for publication.

Best regards,

Reviewer 1 comment: The authors reported HSP contribution in RA pathogenesis.

The article was interesting and useful to understand the roles of HSP in RA, however, the followings are needed to publish in the Journal.

I feel strange the expression of 'the body's' in the text.

I hardly understand 'neurogenic inflammation'. Do you have another term instead of 'neurogenic'?

Line 57: named synovium and Line 60: synovial membrane, These gave me unfamiliar meaning.

Authors’ Reply: Thank you very much for the constructive and positive comments regarding our manuscript. As for the word ‘the Body’s. We deleted the word body’s in the sentences that actually do not need to have this word included. Regarding the word synovium and synovial membrane they both have the same meaning therefore we used only the term synovium, in order not to mislead the readers. However, regarding the term neurogenic inflammation this is very essential in our manuscript and our work since this auto-immune disorder has been linked to a neurogenic inflammation which is a pathophysiological process that includes a complex biological response of the immune, vascular and neural systems. These are some articles that use the term neurogenic inflammation: 1. Sun WH, Dai SP. Tackling Pain Associated with Rheumatoid Arthritis: Proton-Sensing Receptors. Adv Exp Med Biol. 2018;1099:49-64. 2. Pan B, Zhang Z, Chao D, Hogan QH. Dorsal Root Ganglion Field Stimulation Prevents Inflammation and Joint Damage in a Rat Model of Rheumatoid Arthritis. Neuromodulation. 2018 Apr;21(3):247-253. 3. Chiu IM, von Hehn CA, Woolf CJ. Neurogenic inflammation and the peripheral nervous system in host defense and immunopathology. Nat Neurosci. 2012 Jul 26;15(8):1063-7.

Reviewer 2 Report

The manuscript by Fouani et al. represents a review article and is focused on heat shock proteins alterations in rheumatoid arthritis. In my opinion, this manuscript is timely, is well written, well-illustrated and easy to follow. I have a couple of suggestions only, which I believe will improve the presentation:

- Within current version the conclusions section misses the ideas of the authors on the most promising (in their opinion) targets and possible applications for therapeutic interventions;

- The fonts within the figures have potential for the enlargement, which will make the style for illustrations more friendly for the readers.

Author Response

Subject: Manuscript ID: ijms-1621218 “Heat Shock Proteins Alterations in Rheumatoid Arthritis”

Dear Editor,

Thank you for giving us the opportunity to resubmit our manuscript after the peer review process. The comments of the reviewers were very appropriate and we modified the text of the manuscript and the figures in accordance with the reviewers’ requests. All new or modified parts are visible in red font. Below this message you will find the response to the reviewers’ comments.

We hope the manuscript is now acceptable for publication.

Best regards

Reviewer 2 comment: The manuscript by Fouani et al. represents a review article and is focused on heat shock proteins alterations in rheumatoid arthritis. In my opinion, this manuscript is timely, is well written, well-illustrated and easy to follow. I have a couple of suggestions only, which I believe will improve the presentation:

- Within current version the conclusions section misses the ideas of the authors on the most promising (in their opinion) targets and possible applications for therapeutic interventions;

- The fonts within the figures have potential for the enlargement, which will make the style for illustrations more friendly for the readers.

Authors’ Reply: Thank you for your very appropriate comment that give us the opportunity to improve the quality of the figures. As we checked the submitted draft of the review, it seemed that the figures attached were blurry and not clear, we do not know why did this happen though the original figures are very clear. However, we re-attached them again with better resolution. We hope that this version is better. Regarding the conclusion section, we added a part that in our opinion tackles more the promising therapeutic intervention of heat shock proteins.